

# Predicting cervical cancer risk probabilities using advanced H20 AutoML and local interpretable model-agnostic explanation techniques

Sashikanta Prusty[1], Srikanta Patnaik[2], Sujit Kumar Dash[3],
Sushree Gayatri Priyadarsini Prusty[1], Jyotirmayee Rautaray[4] and
Ghanashyam Sahoo[5]

[1] Department of Computer Science and Engineering, Siksha O Anusandhan University Institute of Technical Education and Research, Bhubaneswar, Odisha, India
[2] Director of IIMT, Interscience Institute of Management and Technology, Bhubaneswar, Odisha, India
[3] P & IT, Biju Pattanaik University of Technology, Rourkela, Odisha, India
[4] Department of Computer Science, Odisha University of Technology and Research, Bhubaneswar, Odisha, India
[5] Department of Computer Science and Engineering, GITA Autonomous College, Bhubaneswaer, Odisha, India

Corresponding author
Sashikanta Prusty,
sashi.prusty79@gmail.com

## ABSTRACT

**Background.** Cancer is positioned as a major disease, particularly for middle-aged people, which remains a global concern that can develop in the form of abnormal growth of body cells at any place in the human body. Cervical cancer, often known as cervix cancer, is cancer present in the female cervix. In the area where the endocervix (upper two-thirds of the cervix) and ectocervix (lower third of the cervix) meet, the majority of cervical cancers begin. Despite an influx of people entering the healthcare industry, the demand for machine learning (ML) specialists has recently outpaced the supply. To close the gap, user-friendly applications, such as H2O, have made significant progress these days. However, traditional ML techniques handle each stage of the process separately; whereas H2O AutoML can automate a major portion of the ML workflow, such as automatic training and tuning of multiple models within a user-defined timeframe.

**Methods.** Thus, novel H2O AutoML with local interpretable model-agnostic explanations (LIME) techniques have been proposed in this research work that enhance the predictability of an ML model in a user-defined timeframe. We herein collected the cervical cancer dataset from the freely available Kaggle repository for our research work. The Stacked Ensembles approach, on the other hand, will automatically train H2O models to create a highly predictive ensemble model that will outperform the AutoML Leaderboard in most instances. The novelty of this research is aimed at training the best model using the AutoML technique that helps in reducing the human effort over traditional ML techniques in less amount of time. Additionally, LIME has been implemented over the H2O AutoML model, to uncover black boxes and to explain every individual prediction in our model. We have evaluated our model performance using the findprediction() function on three different idx values (*i.e.*, 100, 120, and 150) to find the prediction probabilities of two classes for each feature. These experiments

have been done in Lenovo core i7 NVidia GeForce 860M GPU laptop in Windows 10 operating system using Python 3.8.3 software on Jupyter 6.4.3 platform.

**Results**. The proposed model resulted in the prediction probabilities depending on the features as 87%, 95%, and 87% for class '0' and 13%, 5%, and 13% for class '1' when idx_value=100, 120, and 150 for the first case; 100% for class '0' and 0% for class '1', when idx_value= 10, 12, and 15 respectively. Additionally, a comparative analysis has been drawn where our proposed model outperforms previous results found in cervical cancer research.

# INTRODUCTION

Cervical cancer is the fourth most prevalent cancer among women worldwide, and the second most common cancer among women in developing countries. World Health Organization (WHO) estimated 570,000 new cases globally in the year 2018. Low- and middle- income countries (LMICs) accounted for over 90% of the 311,000 deaths worldwide in 2018 (*World Health Organization, 2024*). Over 85% of these deaths took place in low- and middle-income nations. Cervical cancer accounts for around 6–29% of all malignancies in women in India. Cervical cancer incidence rates increase rapidly, with the highest rate of 23.07/100,000 in Mizoram and the lowest rate of 4.91/100,000 in the Dibrugarh district in India (*Saleem & Bhattacharya, 2021*). Cervical cancer is related to a young age at marriage, several sexual partners, multiple pregnancies, poor genital hygiene, malnutrition, the use of oral contraceptives, and a lack of awareness. Furthermore, India has the highest (age-standardized) cervical cancer incidence rate in South Asia, with 22 cases per 100,000 women per year (estimations for 2012), compared to 19.2 in Bangladesh, 13 in Sri Lanka, and 2.8 in Iran.

Human papillomavirus (HPV) infection is responsible for nearly all occurrences of cervical cancer (*Yuan et al., 2021*; *Kjaer et al., 2021*). There are around 100 different types of HPV, with at least 14 of them causing cancer. HPV is a virus family that is one of the main causes of sexually transmitted infections in both men and women without having any clinical symptoms. Around the world, HPV having types 16 and 18 are responsible for 70% of cervical malignancies and precancerous lesions. Cervical cancer is caused by HPV infection acquired through sexual contact (*Colombo et al., 2021*; *Bouvard et al., 2021*; *Falcaro et al., 2021*). Although, HPV affects the immune system of patients having HIV increases the cervical cancer risk (*Stelzle et al., 2021*).

Due to a lack of resources, and qualified, and trained health workers in impoverished and underdeveloped countries, the output of cervical cancer screening procedures is low. In developed countries like Sub-Saharan Africa (SSA), pre-cancerous lesion prediction and early-stage treatment efficient screening procedures over nearly 93,225 cases were identified in 2012 (*Black & Richmond, 2018*). Infection with the human papillomavirus is the major risk factor for cervical cancer. In low- and lower-middle-income nations,

cancer-causing diseases such as hepatitis and HPV account for roughly 30% of cancer cases (*de Martel et al., 2020*). The Papanicolaou (PAP) smear is the world's most used cervical cancer screening test. During the screening of each patient, the trained cytologists examine hundreds of sub-images using a microscope to identify the abnormal cells. PAP smear slide shows the cervical cell image, which comprises a mixture of red and white blood cells, germs, and the cervical cell cluster (*Lu et al., 2020*).

## Literature review

Despite a spike in people entering the healthcare field, demand for machine learning (ML) professionals has exceeded supply in recent years due to the availability of a larger dataset. To close this gap, significant progress has been made in the development of non-expert-friendly ML software. The development of simple, consistent interfaces to a variety of machine learning algorithms was one of the first steps toward simplifying machine learning (*e.g.*, H2O). Moreover, H2O has made it simple for non-experts to experiment with machine learning, Deep Neural Networks, in particular, have historically been difficult for non-experts to configure appropriately. The H2O AutoML model might be useful in predicting cervical cancer in less time as it provides a simple wrapper function that performs a large number of modeling-related tasks over the complex image. After evaluating the fivefold cross-validation technique over all datasets, it was found that both ML and deep learning (DL) techniques provide comparatively good results, with ResNet-50 achieving an accuracy of 0.9065 and much better performance. Besides that, the conceptual methodologies in ML may have struggled to comprehend and learn these complex diagnostic processes in cancer disorders. A traditional ML model would normally require many lines of code, allowing the model to focus on other aspects of the task, such as data preprocessing, feature engineering, and model deployment. A major drawback of applying the ML process is the requirement of human interaction at each step, and it may be impossible to accurately compare ML and DL models (*Park et al., 2021*; *Prusty, Patnaik & Dash, 2022*).

The application of ML for medical image analysis has several advantages in terms of disease diagnosis. Over convolutional neural networks (CNN), CNN with conditional random field (CNN-CRF) provides a variety of applications for assessing the structure and capturing a picture of the human interior body structure. The field of medical image analysis has profited greatly from machine learning (*Soni & Soni, 2021*).

Cervical cancer is the fourth-highest rate of increase among female diseases. It is one of the illnesses that is threatening the health of women all over the world, and it is difficult to detect any symptoms in the early stages. However, due to several social-behavioral issues, the cervical cancer screening procedure can be impeded. In the field of gynecology, there are currently a limited number of studies focused on identifying cervical cancer based on behavior and machine learning (*Akter et al., 2021*).

As a result of the COVID-19 outbreak, low-income women's breast and cervical cancer screening rates have decreased. Longer delays in screening owing to COVID-19 risk increase cancer outcome disparities (*Ginsburg et al., 2021*; *DeGroff et al., 2021*; *Feldman & Haas, 2021*; *Castanon et al., 2021*).

Low-income women without health insurance can get cancer screenings under the National Breast and Cervical Cancer Early Detection Program (NBCCEDP). Breast cancer and cervical cancer screening, on the other hand, remained more than 50% below the 5-year average among women in rural areas (*Ortiz et al., 2021*).

Traditional machine learning/deep learning techniques necessitate specialist knowledge, and so rely on human efficiency at the moment. Automated machine learning (AutoML) has evolved as a solution for application domain problems such as anomaly detection as well as unstructured constraints (*Miller et al., 2021*; *Kancharla & Raghu Kishore, 2022*).

Local interpretable model-agnostic explanations (LIME) is a common technique for making black box ML algorithms more interpretable and explainable. LIME frequently creates an explanation for a single ML model prediction by learning a smaller interpretable model (*e.g.*, linear classifier) around the prediction by randomly perturbing simulated data around the instance and determining feature importance through feature selection (*Shi et al., 2021*; *Zafar & Khan, 2021*).

TS-MULE is a time series-specific local surrogate model explanation method that extends the LIME methodology. This enhanced LIME uses a variety of techniques to segment and modify time series data (*Schlegel et al., 2021*).

## Objectives

In general, in traditional ML techniques, each stage of the process is handled separately. Although, the ML process follows some basic steps while predicting healthcare datasets such as data processing, feature engineering, feature selection, model selection, hyperparameters optimization, model performance, analysis of the result, and making a prediction. Whereas, AutoML, is an open-source library that automates each step in the ML process from processing the raw dataset to deploying the ML model. AutoML identifies and employs the most appropriate machine learning algorithm for a specific task. First, there is a neural architecture search, which automates neural network design. This makes it easier for AutoML models to find new architectures for situations that require it. The second is transfer learning, in which previously trained models apply their knowledge to fresh data sets. AutoML can use transfer learning to apply existing structures to new problems. The three most basic steps that H2O AutoML follows are (a) Data Preprocessing, (b) Model Generation, and (c) Ensembling, as shown in Fig. 1.

The H2O AutoML interface is designed to have as few parameters as possible, allowing the experts to simply point to their dataset, identify the response column, and optionally specify a time limitation or a maximum number of models for training. Moreover, AutoML makes it easy to provide faster and more accurate results in healthcare fields. H2O AutoML assists in data preprocessing by determining which parameters to tweak and what ranges will allow us to know when random sampling will occur. The advancement of the H2O AutoML technique over the traditional ML process has been described as shown in Fig. 2.

In this work, the H2O AutoML has been proposed to predict cervical cancer incoming section that automates the selection, composition, and parameterization of the ML model. Even though it increases efficiency and processing power to generate results. The parts of the ML process that apply the algorithm to real-world scenarios are automated in

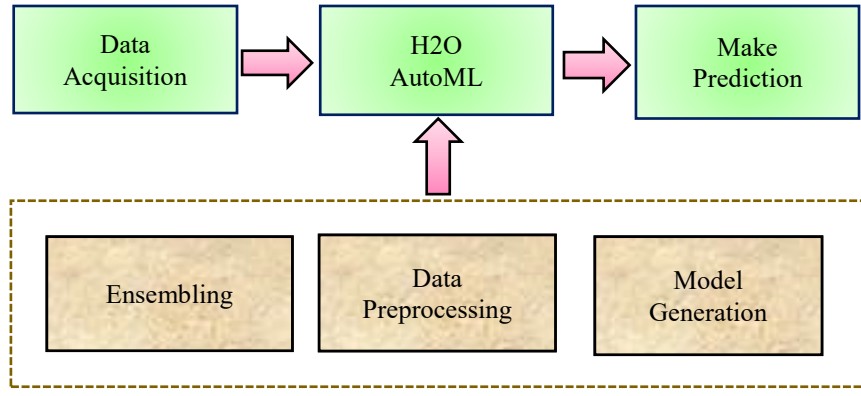

**Figure 1** **H2O AutoML process design.**

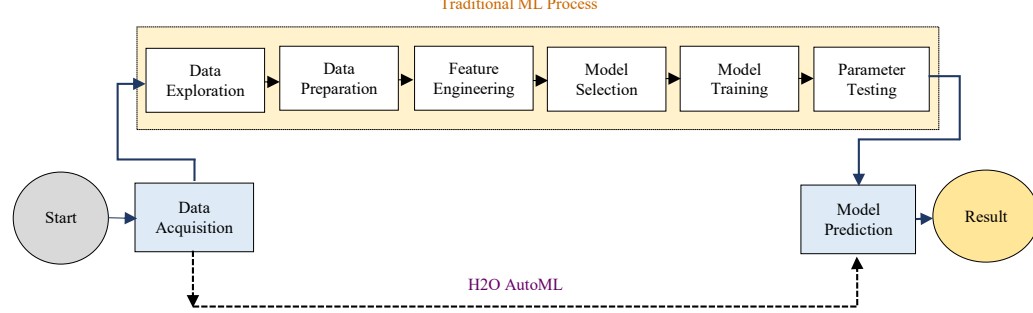

**Figure 2** **Differences between the traditional ML and H2O AutoML process.**

this phase. A human executing this activity would require knowledge of the algorithm's internal logic as well as how it connects to real-world scenarios. It learns about learning and makes decisions that would take too long or need too many resources for people to perform efficiently at scale. Furthermore, H20 AutoML aids research and development in the healthcare sector, as it can evaluate big data sets and derive insights from complex data.

## MATERIALS & METHODS

H2O.AI is an open-source program for analyzing large amounts of data. It allows users to fit thousands of different models to tumour data to find patterns. The software includes a module for creating AutoML models that can be used with programming languages such as Java. Furthermore, AutoML is primarily concerned with data collecting and prediction. The AutoML platform will abstract all of the steps that occur between these two phases.

### Material

In this study, we have collected two separate cervical cancer datasets from the Kaggle repository, which has been publicly available over the Internet. The first dataset contains

858*36 dimensions of patient data, where 858 is defined as the number of patients and 36 as the total number of features (for example, hormonal contraceptives, smokes (year), Dx: cancer, Dx: CIN, and so on). These features are mainly responsible for developing abnormal tissues in the cervix area like possibilities of chlamydia to increase the chances of HPV infection, unprotected sexual intercourse with more than one partner, smoking to produce precancerous changes in the cervix to develop invasive CC, and so on. From these 36, we have dropped two features (*i.e.,* 'time since first diagnosis' and 'time since last diagnosis') as we found more than 80% of missing values. Although, Dx: cancer produces uncertainties about the validity of the prediction, removing it to maintain the integrity of the prediction. Besides that, we also removed three other diagnosis techniques such as 'Hinselmann', 'Schiller', and 'Citology' as they only guide whether cancer is present or not but 'Biopsy' can make a definite diagnosis tool. That is why, we have gone with the 'Biopsy' test to predict the cancer in this research. The second one contains 72 individual patient data and 20 features (for example, behavior_sexualRisk, behavior_eating, behavior_personalHygine, perception_vulnerability, and many more). After all these processes, the dimension of our dataset has been reduced to 858*30, which is undergone into the model for prediction described in a further section.

## Method

Despite the rise in experts in the healthcare field, demand for machine learning professionals has recently exceeded supply. The initial efforts for improving ML were to make it quite easy with the usage of H2O to fill this gap. However, H2O makes it simpler for beginners to evaluate ML models. AutoML, on the other hand, can be useful for the proficient user, freeing them up to concentrate on other areas like data-preprocessing, feature engineering, and deploying models by offering an intuitive wrapper function, which performs a variety of operations that usually involves hundreds of lines during coding. Combining both t H2O and AutoML allows the user to simply point to their dataset, identify the response column, and optionally specify a time constraint or a maximum number of models to train. Instead of a specified time constraint, the user can utilize a performance metric-based termination condition for the AutoML operation. Stacked Ensembles will be automatically trained on a set of individual models to build a highly predictive ensemble model that will be the best performer in the AutoML Leaderboard. The implementation of H2O with AutoML can automate ML activities including automatic training and tuning of multiple models in a user-defined timeframe. Model explainability methods are available in H2O for both AutoML objects and individual models from the leaderboard. With a single function call, a model explanation can be carried out automatically, giving a convenient interface after implementing the LIME application.

The progress of AI in the healthcare industry necessitates that models be explainable to both physicians and users. AI interpretability reveals what is going on inside these systems and aids in the detection of potential problems including information leakage, model bias, robustness, and causality. LIME provides a generic framework for uncovering black boxes in machine learning models.

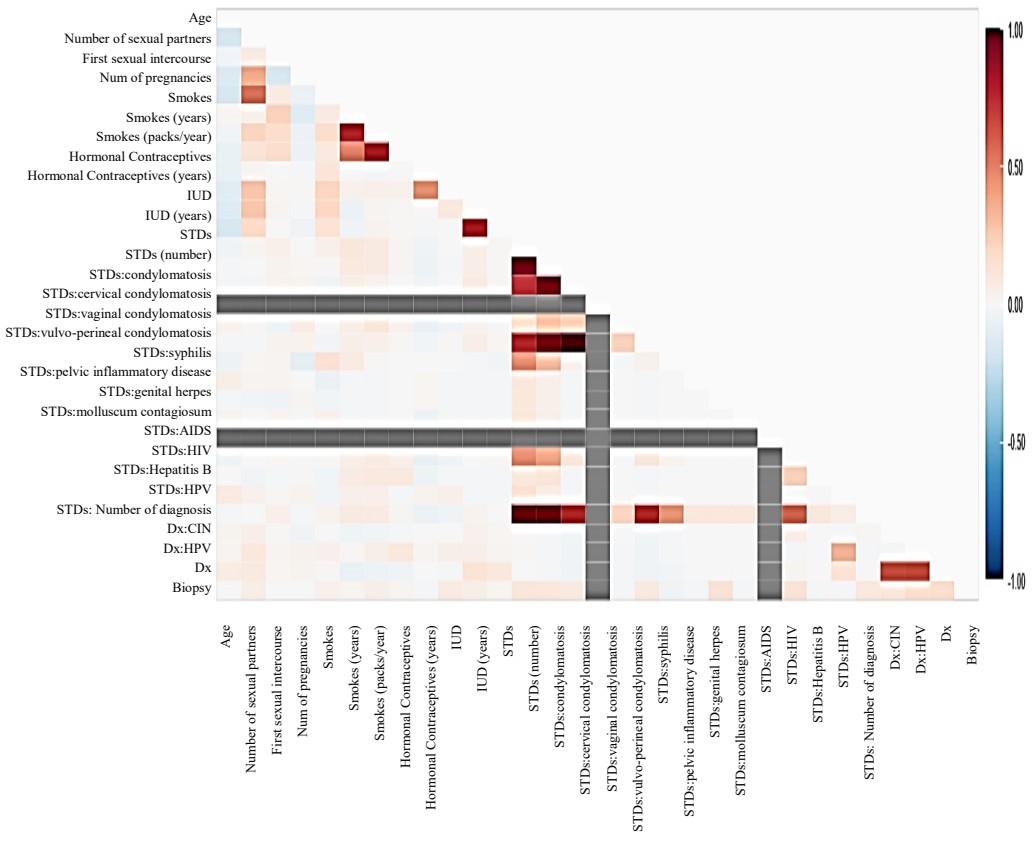

**Figure 3** **We have designed a relationship between 29 features using Pearson correlation coefficient (PCC).**

Before moving forward to method implementation, it is necessary to analyze the dataset such as identifying missing values, finding correlation between features, and many more. For this implementation, the exploratory data analysis (EDA) technique has been used here, which produces a report specifying the relationship between features in the CC dataset. Thus, to define the degree of features dependent on each other, we performed correlation between variables using the Pearson correlation coefficient (Fig. 3), Spearman (Fig. 4), and KendallTau (Fig. 5) in this study.

## Prepare H2O dataframes

Start and connect to the Local H2O cluster:

H2O is a set of Stata utilities for interacting with H2O. This can use these utilities to start or connect to an H2O cluster and have access to H2O's capabilities. The H2O_cluster_version 3.34.0.7 has been connected with our H2O_from_python_Lelin_scenxd cluster using the Python 3.8.8 application on the Windows 10 operating system. This requires an internet browser *i.e.,* Microsoft Edge version 117.0.2045.60 (64-bit) to connect H2O web UI.

Importing Data into H2O:

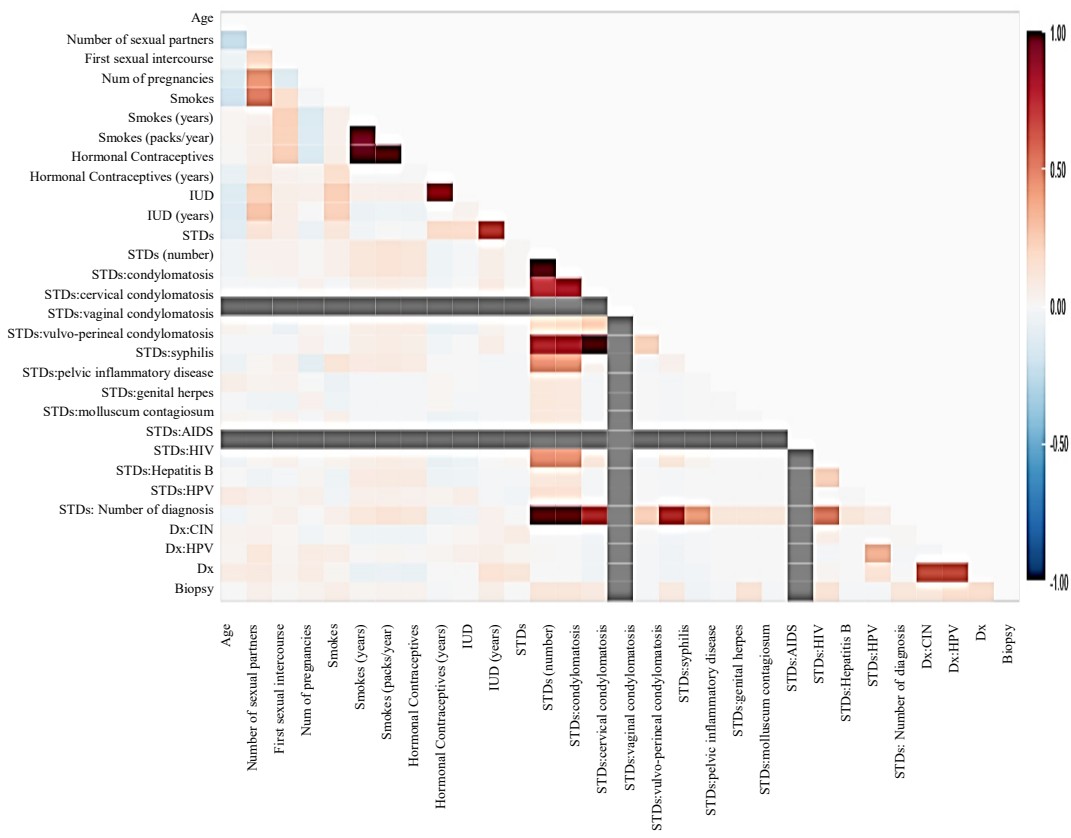

**Figure 4** The relationship between features using Spearman correlation.

The import function is a parallelized reader that retrieves data from the server at a client-specified location. This method of data reading is quick, scalable, and highly optimized. H2O retrieves data from a data store and performs a read operation on it.

Convert pandas Dataframe into H2O data frame:

H2OFrame is identical to Pandas' Dataframe, with the exception that the data is usually not maintained in memory, but rather on a (potentially remote) H2O cluster, easy to handle the data (Fig. 6).

## Data transformation & exploration

The transformation of training data into numeric type can be done using the transform parameter.

Split the H2O data frame into Training and Testing Sets:

The collection of predictors in this Cervical Cancer dataset is all columns, namely 'x', except the Biopsy, which is 'y'.

## Train multiple H2O models

H2O AutoML is configured with max_models as a training parameter that takes the maximum number of models and the validation_frame, which specifies the evaluation

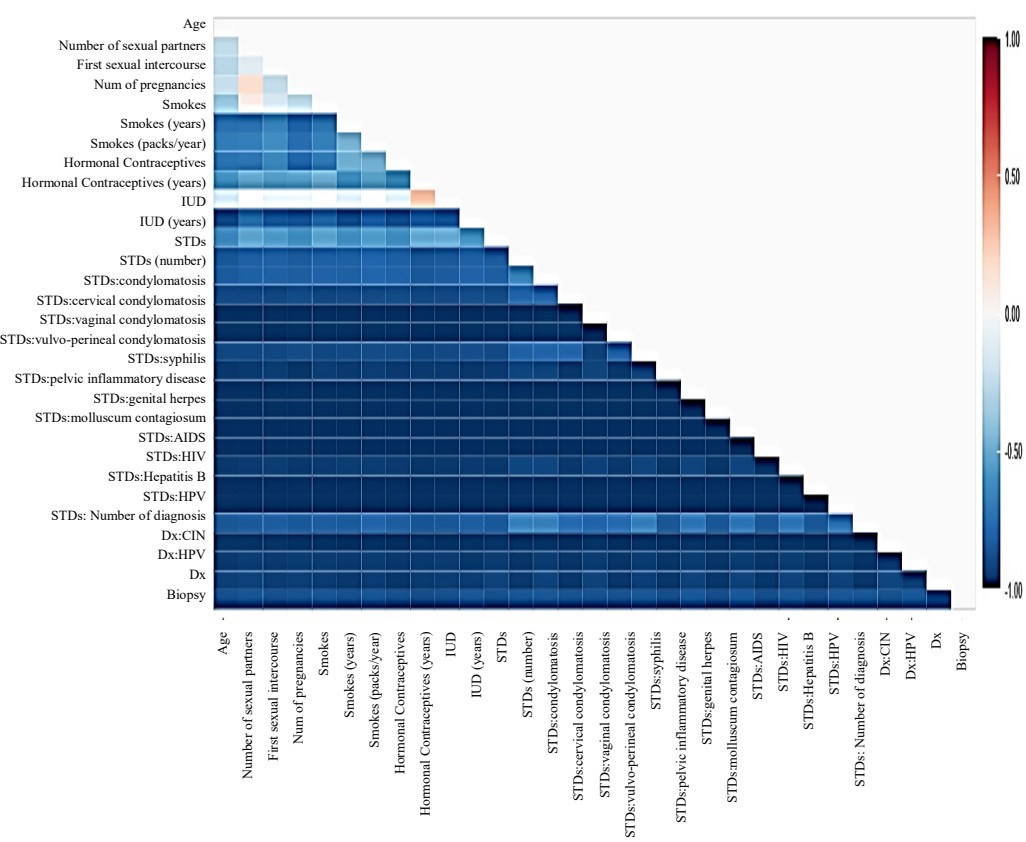

**Figure 5** The relationship between features using KendallTau correlation.

process in H2OFram. However, in this study H2OAutoML () function takes 'max-models =20', metrics as 'ROC AUC', and 'StackedEnsemble' parameters to train our model. The performance score history to represent the timestamp, durations, training_loss, and so on has been designed in Table 1. Rather than that, in this work, H2O AutoML trains and cross-validates twenty models like GBM, default random forest (DRF), extremely randomized forest (XRT), deep learning, XGBoost, and gradient linear machines (GLM). Moreover, it trains stacked ensembles for all the models on the cervical cancer dataset. Other than the twenty models, the GBM creates regression trees progressively according to the dataset characteristics in a fully distributed manner, where each tree is generated in parallel (Table 2). After that, the regression performance can be measured on trained data and cross-validation data using performance metrics. The model's performance can be checked on the training and test set by using confusion_matrix as shown in Table 3.

## H2O AutoML model leaderboard

The H2O Model Explainability interface fully supports AutoML objects. A large number of multi-model comparisons and single-model graphs can be generated automatically with a single call to h2o.explain The leaderboard displays useful and actionable data for each

```
In [30]: h2o.init()
```

Checking whether there is an H2O instance running at http://localhost:54321 ..... not found.
Attempting to start a local H2O server...
; Java HotSpot(TM) 64-Bit Server VM (build 25.311-b11, mixed mode)
  Starting server from C:\Users\Lelin\anaconda3\Lib\site-packages\h2o\backend\bin\h2o.jar
  Ice root: C:\Users\Lelin\AppData\Local\Temp\tmpg11sk9ht
  JVM stdout: C:\Users\Lelin\AppData\Local\Temp\tmpg11sk9ht\h2o_Lelin_started_from_python.out
  JVM stderr: C:\Users\Lelin\AppData\Local\Temp\tmpg11sk9ht\h2o_Lelin_started_from_python.err
  Server is running at http://127.0.0.1:54321
Connecting to H2O server at http://127.0.0.1:54321 ... successful.

| | |
|---|---|
| H2O_cluster_uptime: | 07 secs |
| H2O_cluster_timezone: | America/Los_Angeles |
| H2O_data_parsing_timezone: | UTC |
| H2O_cluster_version: | 3.34.0.7 |
| H2O_cluster_version_age: | 19 days |
| H2O_cluster_name: | H2O_from_python_Lelin_scenxd |
| H2O_cluster_total_nodes: | 1 |
| H2O_cluster_free_memory: | 1.755 Gb |
| H2O_cluster_total_cores: | 0 |
| H2O_cluster_allowed_cores: | 0 |
| H2O_cluster_status: | locked, healthy |
| H2O_connection_url: | http://127.0.0.1:54321 |
| H2O_connection_proxy: | {"http": null, "https": null} |
| H2O_internal_security: | False |
| H2O_API_Extensions: | Amazon S3, Algos, AutoML, Core V3, TargetEncoder, Core V4 |
| Python_version: | 3.8.8 final |

**Figure 6** **Connecting to H2O server using h2o.init().**

**Table 1** **Training Score history using the H2OAutoML technique on first case.**

| Timestamp | Duration | number_of_trees | training_logloss | training_auc | training_pr_auc | training_lift | training_ classification_ error |
|---|---|---|---|---|---|---|---|
| 2023-10-07 11:29:47 | 4.663 s | 0.0 | 1.373910 | 0.500000 | 0.501266 | 1.000000 | 0.498734 |
| 2023-10-07 11:29:47 | 4.703 s | 5.0 | 1.293767 | 0.870937 | 0.862830 | 1.994949 | 0.197468 |
| 2023-10-07 11:29:47 | 4.728 s | 10.0 | 1.198367 | 0.939511 | 0.935237 | 1.994949 | 0.124051 |
| 2023-10-07 11:29:47 | 4.754 s | 15.0 | 1.116737 | 0.948414 | 0.944879 | 1.994949 | 0.112236 |
| 2023-10-07 11:29:47 | 4.779 s | 20.0 | 1.041071 | 0.959716 | 0.956614 | 1.994949 | 0.095359 |
| 2023-10-07 11:29:47 | 4.811 s | 25.0 | 0.963458 | 0.972699 | 0.970629 | 1.994949 | 0.080169 |
| 2023-10-07 11:29:47 | 4.839 s | 30.0 | 0.902979 | 0.979586 | 0.978727 | 1.994949 | 0.064135 |
| 2023-10-07 11:29:47 | 4.856 s | 32.0 | 0.883423 | 0.981902 | 0.980589 | 1.994949 | 0.056540 |

Prusty et al. (2024), *PeerJ Comput. Sci.*, DOI 10.7717/peerj-cs.1916

**Table 2 Performance metrics of twenty different models using H2OAutoML for first case.**

| model_id | roc auc | logloss | aucpr | mean_per_class_error | mse | training_time_ms | predict_time_per_row_ms | algorithm |
|---|---|---|---|---|---|---|---|---|
| GBM_grid_1_AutoML_1_20220110_123655_model_5 | 0.690086 | 0.244167 | 0.162172 | 0.33666 | 0.0629848 | 196 | 0.021187 | GBM |
| GBM_grid_1_AutoML_1_20220110_123655_model_2 | 0.669512 | 0.250431 | 0.146136 | 0.356368 | 0.0634068 | 20,164 | 0.033766 | Deep Learning |
| GBM_grid_1_AutoML_1_20220110_123655_model_3 | 0.658091 | 0.253076 | 0.135011 | 0.377403 | 0.0637181 | 472 | 0.01396 | GBM |
| DeepLearning_grid_2_AutoML_1_20220110_123655_model_1 | 0.654611 | 0.563843 | 0.163702 | 0.390344 | 0.072609 | 397 | 0.015602 | GBM |
| GBM_grid_1_AutoML_1_20220110_123655_model_4 | 0.6519 | 0.253983 | 0.103104 | 0.330718 | 0.0657598 | 267 | 0.014876 | GBM |
| GBM_1_AutoML_1_20220110_123655 | 0.650804 | 0.253299 | 0.103425 | 0.376058 | 0.0657407 | 275 | 0.009541 | XGBoost |
| GBM_4_AutoML_1_20220110_123655 | 0.644401 | 0.291367 | 0.158755 | 0.372327 | 0.0641633 | 561 | 0.023304 | DRF |
| GBM_grid_1_AutoML_1_20220110_123655_model_1 | 0.631711 | 0.249374 | 0.0874604 | 0.406764 | 0.0647613 | 275 | 0.009143 | XGBoost |
| DeepLearning_grid_3_AutoML_1_20220110_123655_model_2 | 0.631403 | 1.13684 | 0.167609 | 0.393363 | 0.0868137 | 289 | 0.008156 | XGBoost |
| GBM_3_AutoML_1_20220110_123655 | 0.625635 | 0.306683 | 0.149494 | 0.405207 | 0.0651245 | 303 | 0.014975 | GBM |
| DeepLearning_grid_1_AutoML_1_20220110_123655_model_1 | 0.622289 | 0.606564 | 0.108121 | 0.400977 | 0.107165 | 455 | 0.023235 | GBM |
| DRF_1_AutoML_1_20220110_123655 | 0.620635 | 0.553798 | 0.160526 | 0.442816 | 0.0653268 | 511 | 0.01651 | DRF |
| DeepLearning_grid_3_AutoML_1_20220110_123655_model_1 | 0.605426 | 0.866563 | 0.100175 | 0.434106 | 0.0867275 | 326 | 0.013188 | Deep Learning |
| GBM_2_AutoML_1_20220110_123655 | 0.605099 | 0.299854 | 0.151125 | 0.42903 | 0.0644403 | 289 | 0.015554 | GBM |
| XRT_1_AutoML_1_20220110_123655 | 0.585352 | 0.278178 | 0.132334 | 0.446201 | 0.0659885 | 185 | 0.009838 | GLM |
| GBM_5_AutoML_1_20220110_123655 | 0.558856 | 0.313753 | 0.111221 | 0.447412 | 0.0660184 | 15,714 | 0.020604 | Deep Learning |
| DeepLearning_grid_2_AutoML_1_20220110_123655_model_2 | 0.549223 | 1.03666 | 0.104306 | 0.433626 | 0.0762662 | 23,706 | 0.01474 | Deep Learning |
| GLM_1_AutoML_1_20220110_123655 | 0.54332 | 0.247843 | 0.11052 | 0.427453 | 0.0638368 | 302 | 0.00703 | XGBoost |
| DeepLearning_grid_1_AutoML_1_20220110_123655_model_2 | 0.536168 | 1.6599 | 0.0980909 | 0.420185 | 0.0970303 | 189 | 0.006945 | XGBoost |
| DeepLearning_1_AutoML_1_20220110_123655 | 0.531053 | 0.409423 | 0.079844 | 0.433376 | 0.0869002 | 85 | 0.006305 | XGBoost |

**Table 3 Confusion matrix design (Actual *vs* predicted class) on two datasets.**

| | | First case/second case | | First case/second case | |
| | | 0 | 1 | Error | Rate |
|---|---|---|---|---|---|
| 0 | 0 | 211.0/13.0 | 1.0/0.0 | 0.0047/0.0 | (1.0/212.0)/(0.0/13.0) |
| 1 | 1 | 9.0/4.0 | 2.0/2.0 | 0.8182/0.6667 | (9.0/11.0)/(4.0/6.0) |
| 2 | Total | 220.0/17.0 | 3.0/2.0 | 0.0448/0.2105 | (11.0/223.0)/(4.0/19.0) |

**Table 4 Preview of the combined prediction results using H2O AutoML Leaderboard Exploration on two cervical cancer test datasets.**

| First Case | | | | Second Case | | | |
| Biopsy | predict | p0 | p1 | ca_cervix | predict | p0 | p1 |
|---|---|---|---|---|---|---|---|
| 0 | 0 | 0.984997 | 0.0150032 | 1 | 0 | 3.81E−12 | 1 |
| 0 | 0 | 0.991045 | 0.00895493 | 1 | 0 | 1.63E−11 | 1 |
| 0 | 0 | 0.989133 | 0.0108669 | 1 | 1 | 6.08E−17 | 1 |
| 0 | 0 | 0.975303 | 0.0246971 | 1 | 1 | 3.50E−14 | 1 |
| 0 | 1 | 0.922194 | 0.0778059 | 1 | 1 | 5.99E−15 | 1 |
| 0 | 0 | 0.983873 | 0.0161273 | 1 | 1 | 3.65E−17 | 1 |
| 0 | 0 | 0.971292 | 0.0287078 | 0 | 0 | 0.001082576 | 0.998917424 |
| 0 | 1 | 0.890686 | 0.109314 | 0 | 0 | 0.999999999 | 1.18E−09 |
| 0 | 0 | 0.990004 | 0.00999598 | 0 | 0 | 0.999978515 | 2.15E−05 |
| 1 | 1 | 0.786601 | 0.213399 | 0 | 0 | 0.116430347 | 0.883569653 |

model built in the AutoML run, such as model performance, training time, and per-row prediction speed, which is ordered according to user preferences.

The leaderboard displays the metrics for each model. The leaderboard displays 5-fold cross-validated metrics by default (depending on the H2OAutoML parameters) when an H2OAutoML object is provided; otherwise, metrics computed on the frame are displayed. The predict () on the H2OAutoML predict Leaderboard using test scores. The combined scores of prediction values on the test set have been represented in a single data frame as previewed in Table 4 (contains 'p0' and 'p1' as prediction probabilities). From these experiments, a total of 222 test values have been generated, where each biopsy contains a prediction score value and a total of 18 test values have been generated, where each ca_cervix contains a prediction score value In the model evaluation process, the 'nfolds' parameter is used for changing the number of folds in the leaderboard (here default nfold = 5). During the run, AutoML trains multiple Stacked Ensemble models (unless ensembles are disabled using exclude_algos). We divided the AutoML model training into "model groups" with various levels of priority. We train (maximum) two additional Stacked Ensembles using the pre-existing models once each group is finished and at the final stage of the AutoML cycle. Although, to rank, the models (the second column of the leaderboard), a default metric has been implemented that is based on the problem category. However, 'ROC AUC' specifies the statistic for binary classification issues, and 'mean_per_class_error' shows the metric for multiclass classification problems.

## H2O making prediction

In this study, AutoML employs leaderboard expansion and the five-fold cross-validation approach on the CC dataset to produce precise models to predict with their metrics for cervical cancer. When used with AutoML, the leaderboard_frame prospect specifies which data frame will be used to assess and promote models on the leaderboard. With AutoML, the predict () function provides predictions on the run's leader model. Although some rows fail, the results are arranged in the same order as the data was loaded. This graph depicts the relationship between the model's predictions. The frequency of identical predictions is used to classify the data. The similarity of all twenty models is sorted accordingly. Models that can be interpreted, such as GAM, GLM, and RuleFit, are marked in red language in Fig. 7. To investigate and further analyze, the H2O AutoML models use the model explainability interface that will help to decide which model to choose for predicting cervical cancer in advance. The model leader explains H2O models (a total of 20 models by default), and from there the confusion matrix (CM) for GBM_grid_1_AutoML_1 has been described in Table 3, where the output of both the predicted class and actual class.

## H2O: LIME

LIME has a lot of benefits and is adaptable to a variety of ML models due to its faster computation rate (*Molnar, 2019*). The effect of LIME explanation based on human decision-making, where it first examined the outcomes for every cluster, which specifies LIME might perform better in decision support. LIME uses a normal distribution to approximate the original distribution of numerical features and an exponential kernel, with a width corresponding to the square root of the number of features. Superpixels are produced by LIME to classify images, as a result of over-segmenting an image. Superpixels have greater alignment with image edges than rectangular image patches and can hold more data than pixels for the main prediction (*Ahsan et al., 2021*). A total of 10 observations from the test set that had a prediction probability for both classes of more than 80% and an explanation fit of 0.066 were chosen as the LIME assessment instances (*Kumarakulasinghe et al., 2020*). LIME localizes the model as a logistic or linear model and repeats the process hundreds of times. It outputs the most significant features for the local models. LIME has been used to interpret complex models such as Neural Networks, Random Forests, and Ensemble ML. However, an ML model is treated as a "black box" with raw data and certain outputs by generalized explainable AI algorithms (*Sumit, 2019*). Instead of training global intermediate models, LIME trains local simulated models on modified inputs to identify the statistical relationship between variables and model prediction (*Sumit, 2019*). It offers visualization as well as an explanation of an instance through a comprehensible representation. The explanation is measured after evaluating a model behavior close to an instance using local intermediary models, which can be either decision trees or linear regressions as shown in Eq. (1).

$$explanation(x) = arg\ min g \epsilon GL(f, g, \pi x) + \Omega(g) \tag{1}$$

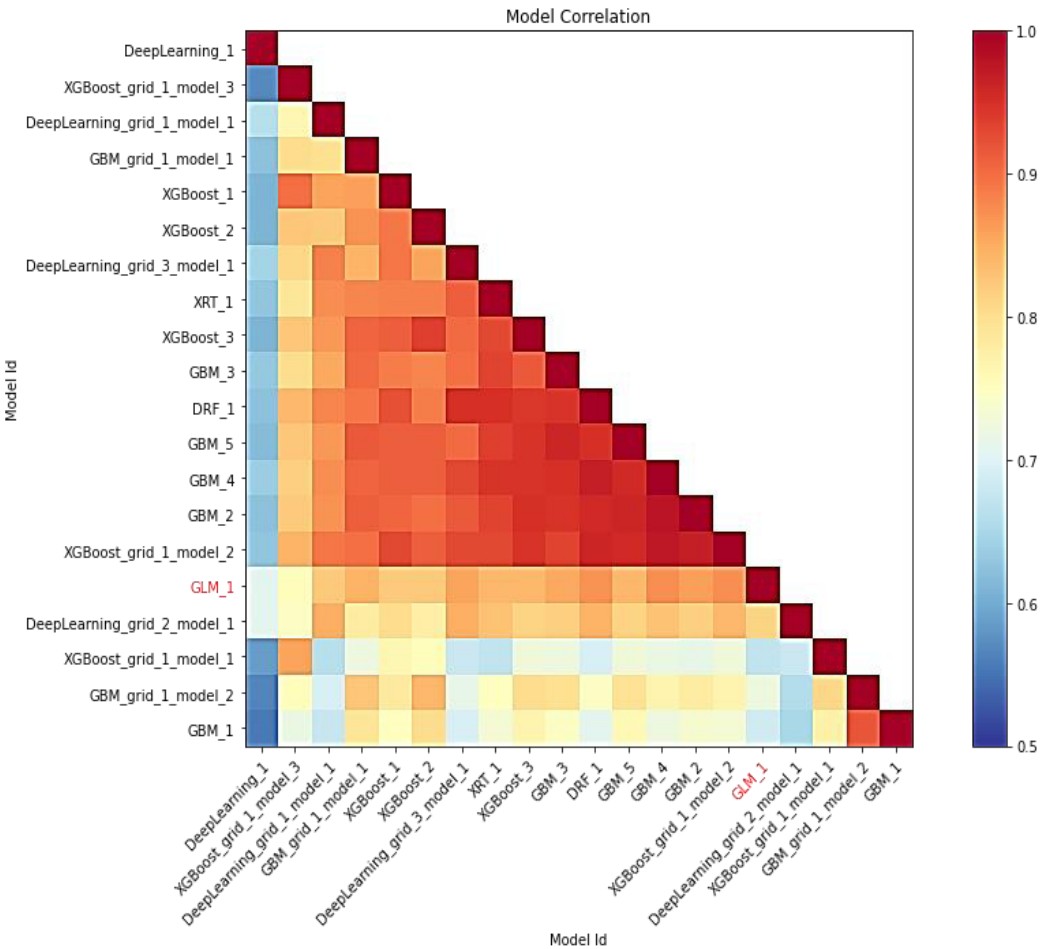

**Figure 7** Depicting the relative correlation performance of 20 default models.

In this case, 'x' is explained on loss function 'L' using the maximum value of $(f, g, \pi x)$, where $\Omega(g)$ is model complexity, 'f' is the black-box model, and 'G' is the family of explanations. 'L' defines how close the explanation result is to the model prediction.

LIME permutes the given observation to compute the distance of similarity between the original and permuted data. In this case, features are passed in h20 frame format, which is further converted into NumPy format for further processing. After that, an explain instance function takes instances in NumPy format, prediction (), and a total number of features as parameters. The findprediction () function creates a data frame object, and h2o frames for the prediction object in a three-column format, where the first column is the prediction class and the other two are their prediction probabilities. This process includes three basic steps:

- First, disregard the training data and consider a black box model in which we provide the input data. The predictions for the model are produced by the black box model. Our goal is to comprehend the reasoning behind the ML model's particular prediction.

**Table 5   ModelMetricsBinomial using H2OAutoML.**

| Metrics | Train data | | Cross-validation data | |
|---|---|---|---|---|
| | First case | Second case | First case | Second case |
| *MSE* | 0.344 | 4.47e−21 | 0.637 | 0.181 |
| *LogLoss* | 0.941 | 1.36e−11 | 0.253 | 0.999 |
| *Mean per class Error* | 0.071 | 0.0 | 0.350 | 0.929 |
| *ROC AUC* | 0.974 | 1.0 | 0.658 | 0.975 |
| *AUCPR* | 0.973 | 1.0 | 0.135 | 0.94 |
| *Gini* | 0.949 | 1.0 | 0.316 | 0.95 |
| | **Test Data** | | | |
| | First Case | | Second case | |
| *MCC* | 0.85 | | 0.93 | |
| *F1 score* | 0.923 | | 0.997 | |
| *accuracy* | 0.978 | | 1.0 | |
| *Sensitivity* | 0.927 | | 0.939 | |
| *Specificity* | 0.975 | | 0.987 | |
| *Precision* | 0.985 | | 0.946 | |
| *NPV* | 0.842 | | 0.897 | |

- LIME is now put into practice. It investigates the changes that occur to the results when we alter the data that an ML model is supplied.
- LIME creates a new dataset made up of permuted data and appropriate black box model predictions. This subsequently trains a comprehensible model on this new dataset.

In this study, we have mainly focused on prediction probabilities. Additionally, we have experimented with our model with three different instance values for our research inside the 'test_numpy' parameter (*i.e.,* 'IDX' values as 100, 120, and 150). These successfully predict the probabilities of class either '0' or '1' in three different HTML files as shown in Figs. 8, 9 and 10. Furthermore, the ML model is applied to predict the outcome of that permuted cancer data. After that, it fits the best appropriate model on the permuted data to explain the outcome, by applying the weights to the original observation. And finally, the feature weights have been used on the cervical cancer dataset that emerged to explain local behavior.

## Performance metrics

H2O AutoML includes several measures for evaluating both supervised and unsupervised models. Only supervised learning models are covered by the metrics in this article, which change depending on the model type. The performance score of both training and cross-validation data for our model has been designed using some common metrics as followed in Table 5. Furthermore, Table 5 contains the performance results on the test set using well-known classification metrics ((for example: MCC, accuracy, precision, specificity, and many more) for two cervical cancer datasets. The result shows that our proposed H2O AutoML model gives significant scores.

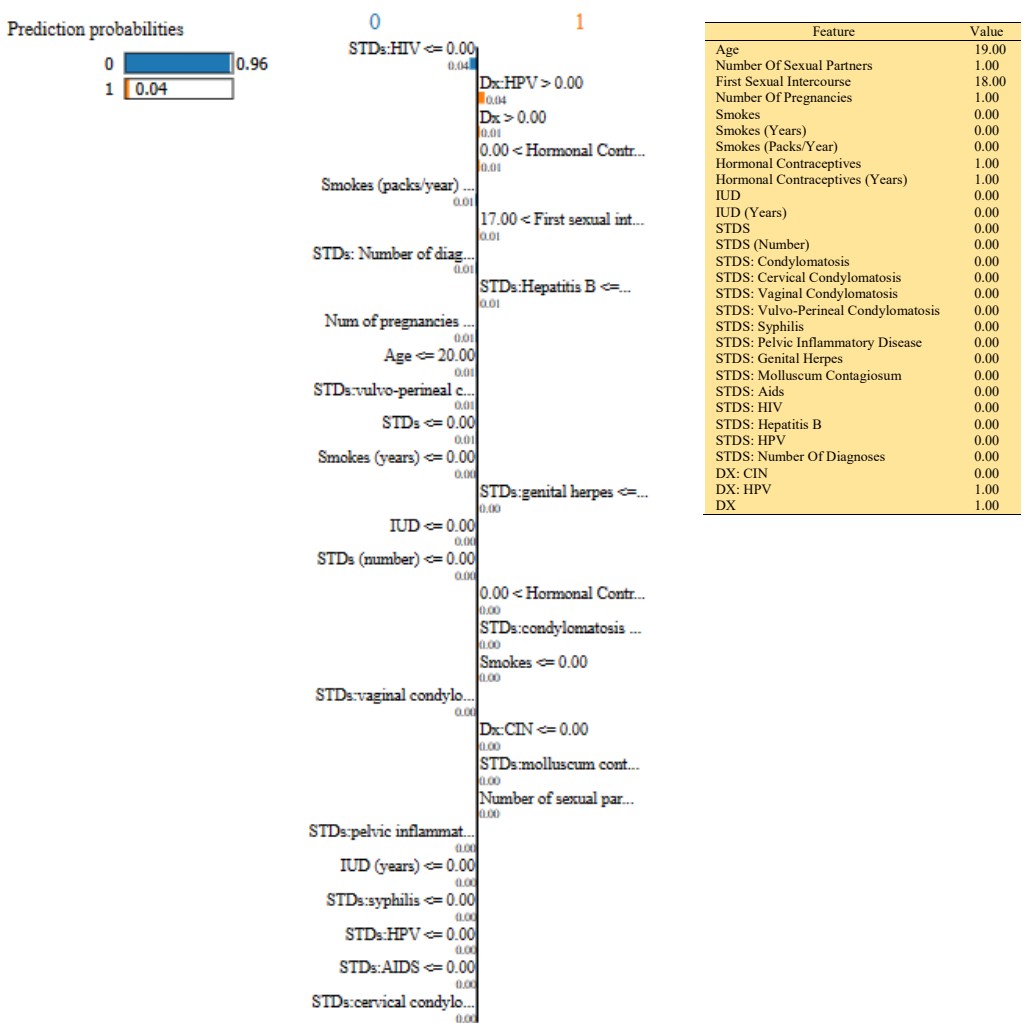

**Figure 8** Based on the H2O AutoML-LIME model prediction probabilities on 29 features for both class '0' and '1' when IDX = 100 in the second case, it appears that there is a higher likelihood of belonging to class '1'.

## Mean-squared error

The mean-squared error (MSE) measure determines the average of the squared errors or deviations. To eliminate any negative signals, MSE squares the distances between the points and the regression line. MSE takes into account the predictor's variation as well as its bias.

$$MSE = \frac{\sum_{i=1}^{N} (\hat{y}_i - y_i)^2}{N} \qquad (2)$$

Where, N - Total number of rows. y - Actual class value. $\hat{y}_i$ - Predicted class value ($\hat{y}_i = \frac{1}{1+e^{-z}}$, as Sigmoid function).

## Log loss

A binomial or multinomial classifier's performance can be evaluated using the logarithmic loss metric. Unlike ROC AUC, which assesses a model's ability to categorize a binary target,

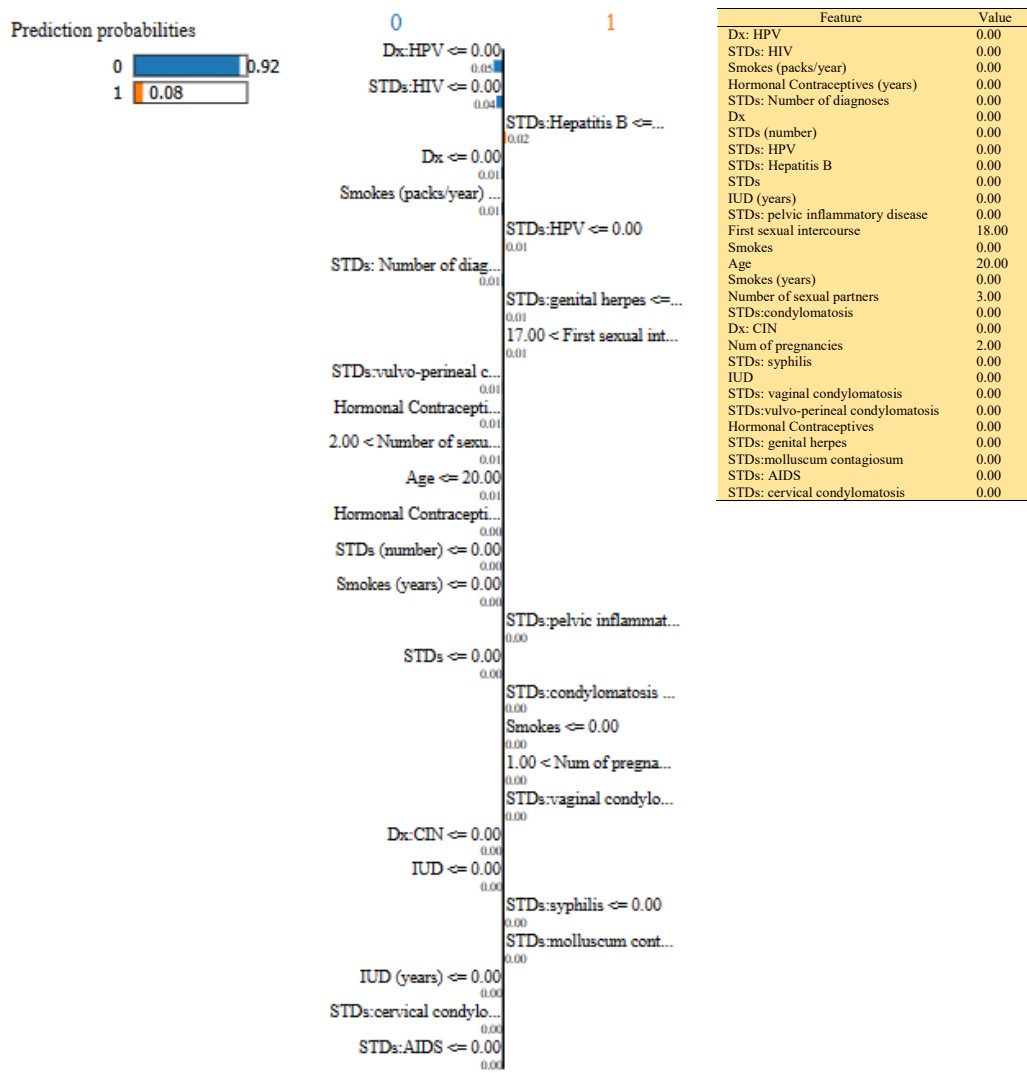

**Figure 9** Based on the H2O AutoML-LIME model prediction probabilities on 29 features for both class '0' and '1' when idx = 120 in the second case, it appears that there is a higher likelihood of belonging to class '1'.

Log loss assesses how near a model's predicted values are to the actual target value.

$$log\ loss = -\frac{1}{N}\sum_{i=1}^{N}(log(p_i)), where\ p_i = prediction\ values(i.e.,\ p_0\ and\ p_1) \qquad (3)$$

Here, $log(p_i)$, defines the probability of a true class (where, $(p_i)$ = probability of class '1' and 1- $(p_i)$ = probability of class '0').

## Area under the ROC curve

This model metric is used to assess how well a binary classification model can differentiate between true and false positives within a graph. Although, this helps to evaluate how well our model decides on actual and predicted values. A perfect classifier has an area under

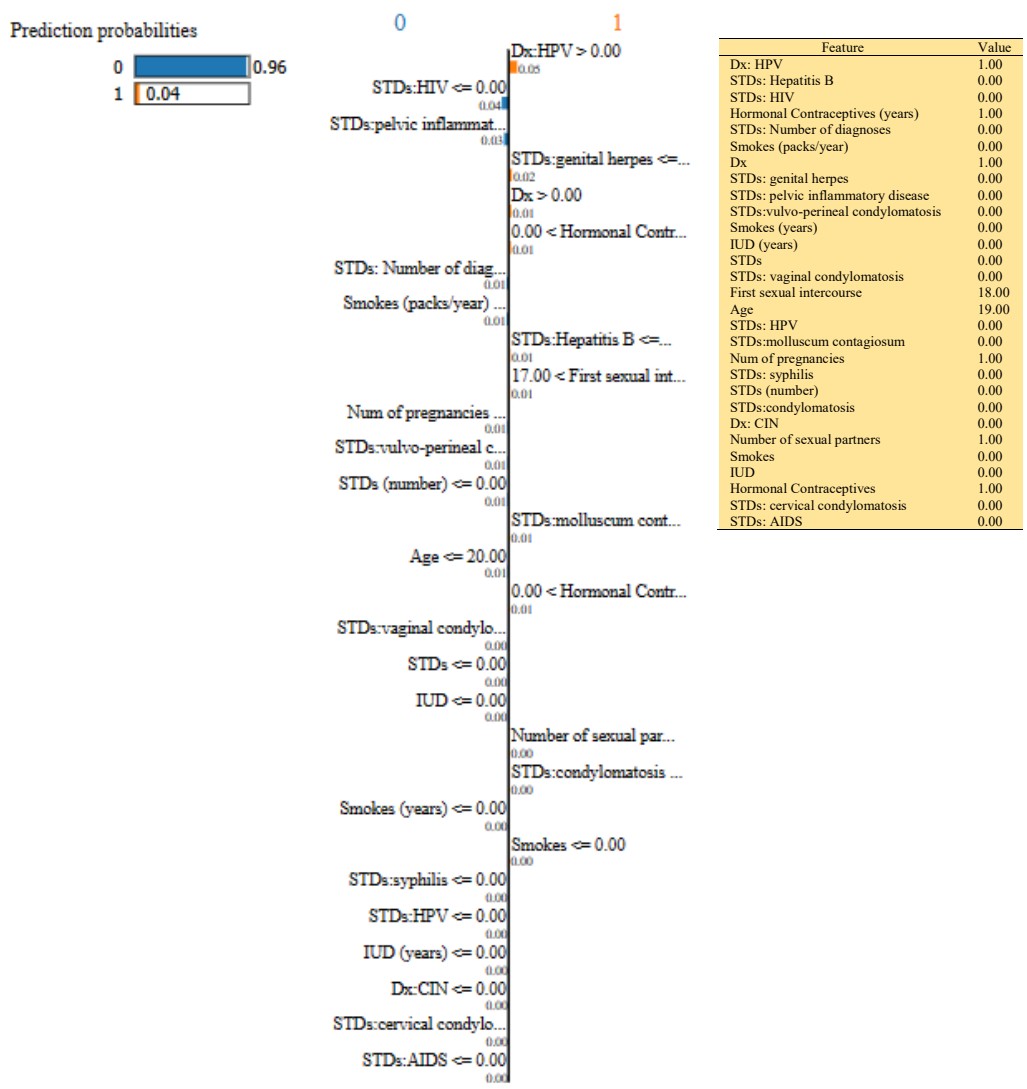

**Figure 10** Based on the H2O AutoML-LIME model prediction probabilities on 29 features for both class '0' and '1' when IDX = 150 in the second case, it appears that there is a higher likelihood of belonging to class '1'.

the ROC curve (ROC AUC) of '1', while a mediocre classifier has an ROC AUC of '0.5'. H2O approximates the area under the ROC curve using the trapezoidal rule. Because a high proportion of True Negatives can lead the ROC AUC to appear inflated, area under the precision-recall curve (ROC AUCPR) can be utilized for an imbalanced binary target in this case. So, ROC-ROC AUC can be mathematically represented as:

$$P(y_1 > y_0) = P(y_1 - y_0 > 0) \tag{4}$$

Where, $y_1 = positive\ samples$ and $y_0 = negative\ samples$

## Area under the precision-recall curve

This model measure assesses a binary classification model's ability to distinguish between precision–recall pairs of points. Different thresholds on a probabilistic or other continuous-output classifier are used to achieve these values. ROC AUC calculates the area under the ROC curve, while ROC AUCPR calculates the area under the Precision-Recall curve. The precision is used to identify the total number of correct predictions where Recall finds correct predictions from all predictions that occurred during the testing phase.

## Gini coefficient

This statistic is frequently used to compare and evaluate the quality of different models and their prediction potential. Gini is the ratio of the area of the ROC curve to the area of the diagonal line and the area of the triangle.

$$Gini = 2 * (AUC - 1). \tag{5}$$

## RESULTS

As described, H2O AutoML helps the user, to be more accessible and to extract useful insights from raw data without the need for deploying ML models. This leaderboard demonstrates that the best accuracy comes from a stacked ensemble model. This leaderboard demonstrates that the best accuracy comes from a stacked ensemble model. Moreover, the leaderboard shows the models (default is 20) with metrics, when H2OAutoML is attached with 5-fold cross-validation as described in Table 2. This cervical cancer dataset has not been pre-processed or subjected to any feature engineering. The performance for combined prediction results using H2OAutoML Leaderboard Exploration has been performed on two separate cervical cancer datasets, collected from Kaggle public repository. The predict () function provides predictions over a test set on the leader model from the run. In the first case, predictions are made based on the 'Biopsy' test (*i.e.,* 224 individual patients data out of 858), and in the second case, predictions are made based on ca_cervix (*i.e.,* 18 individual patients data out of 72) to find their respective classes this can be achieved using two prediction probabilities, namely p0 and p1 (Table 4) on the test dataset. In the above analysis, we can state that the H20 AutoML Leaderboard Exploration significantly predicts cervical cancer in both cases. Additionally, LIME has been implemented that delivers human-understandable context and provides greater explainability of the H2O AutoML model, while predicting this cervical cancer dataset. Lime generates an explanation for a particular observation bypassing the explain_instance () function, where our data is in the form of tables, images, and text. Three different prediction probabilities have been carried out in this project, that take different feature values such as age, the number of sexual partners, first sexual intercourse, more specifically the number of pregnancies, and others. Finally, we predicted the cancer class using both the LIME explainer and H20 AutoML model on two separate datasets for three different scenarios (*i.e.,* idx values). The results are displayed in Figs. 8, 9 and 10 for the first case, whereas Figs. 11, 12 and 13 for the second case. These findings show the predicted probabilities scores for class '1' as orange color and for class '0' as blue color respectively.
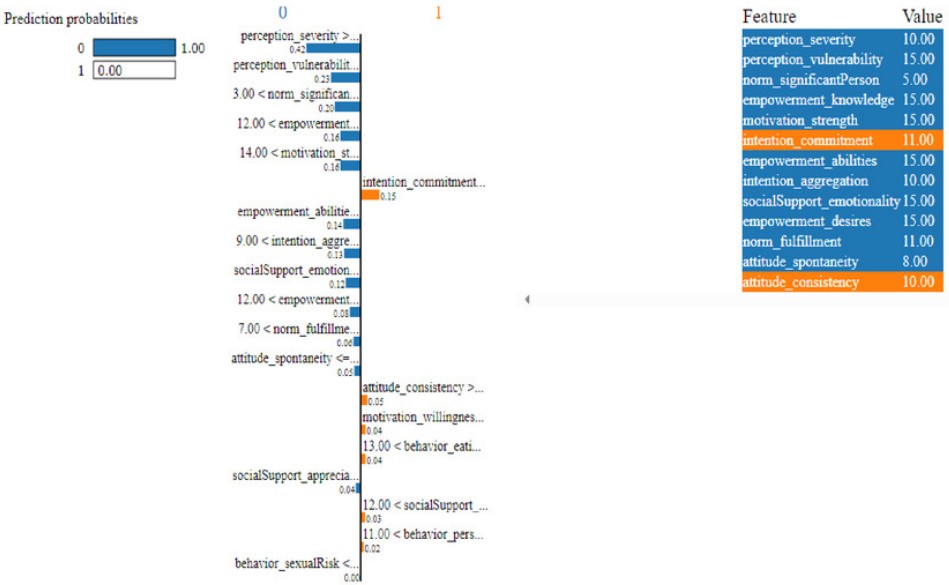

**Figure 11** Based on the H2O AutoML-LIME model prediction probabilities on 19 features for both class '0' and '1' when IDX = 10 in the second case.

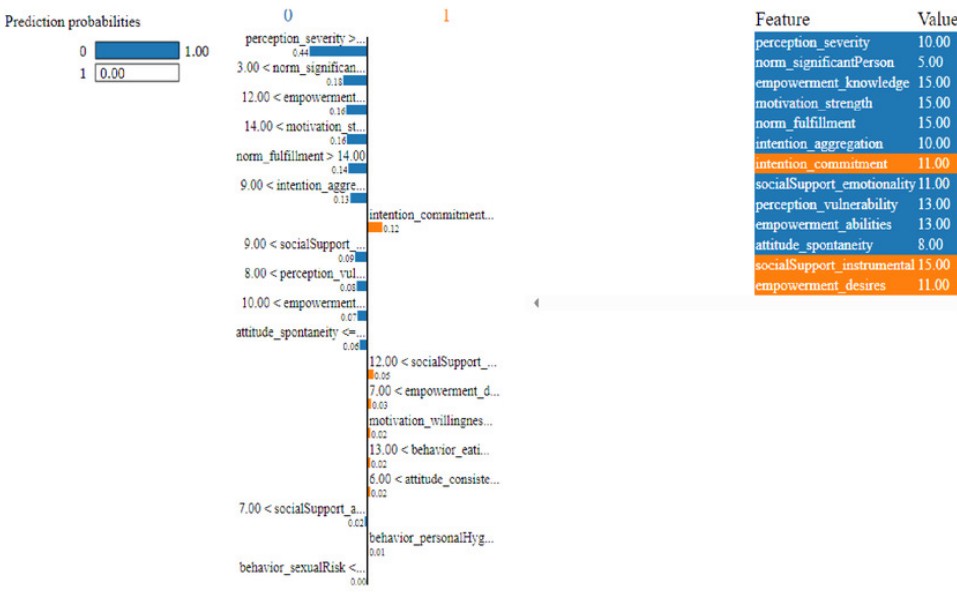

**Figure 12** Based on the H2O AutoML-LIME model prediction probabilities on 19 features for both class '0' and '1' when IDX = 12 in the second case.

Finally, a comparative analysis based on features using the prediction probabilities of class '0' and '1' for three different idx values on two cervical cancer datasets has been designed in Table 6.

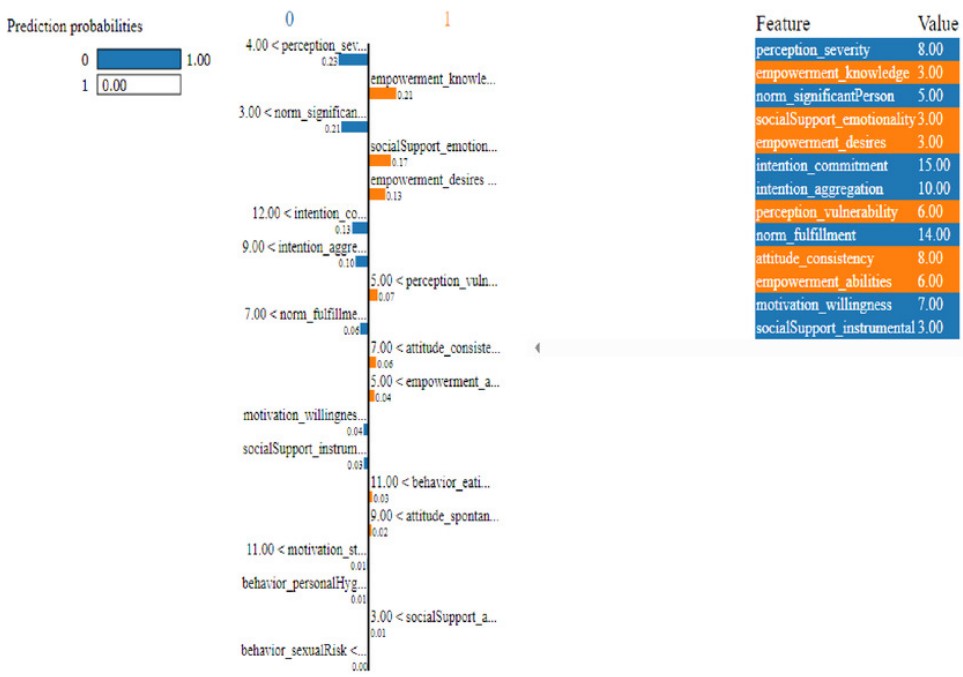

**Figure 13  Based on the H2O AutoML-LIME model prediction probabilities for 19 features for both classes '0' and '1' when IDX = 15 in the second case.**

## DISCUSSION

From the above analysis, we found our proposed method gives the best results in terms of accuracy and minimal processing time with fewer errors as it trains a large number of models at a time on training data. The H2O, on the other hand, provides a faster way to build an ML model on a huge volume of data and also, increases the predictive analytic capabilities. The Java application (JRE 64bit) inside H2O supports multithreading, where the data is distributed parallels across the cluster. These clusters have been stored in an H2O data frame in columnar format. More importantly, H2O AutoML removes the basic steps followed in traditional ML implementation, which greatly reduces the errors and processing time. Although, this technique provides user-friendly interfaces such as model explainability methods to deal with individual or groups of ML models with the help of a single function call.

Gradient boosting machines (GBMs) and balance classes were successfully chosen here for inclusion in the predictive models using AutoML analysis on the CC dataset without affecting prediction accuracy. The H2O AutoML trains a random grid of GBMs, GLMs, and deep neural networks (DNNs) using hyper-parameter space and also tunes every model using cross-validation. Twenty default models were built specifically by AutoML on the leaderboard, and each of them had dramatically improved performance metrics for the algorithms they were paired with. A significant benefit of using GBM_grid_1_AutoML_1_20220110_123655_model_5 in a disease prediction test is

**Table 6 Prediction probabilities of class '0' and '1' using three different idx values on two cervical cancer dataset.**

| | First Case | | | | | |
|---|---|---|---|---|---|---|
| **Features Importance** | **Idx =100** | | **Idx=120** | | **Idx=150** | |
| | **0** | **1** | **0** | **1** | **0** | **1** |
| *Dx: HPV* | | 0.04 | 0.05 | | | 0.05 |
| *STDs: HIV* | 0.04 | | 0.04 | | 0.04 | |
| *Hormonal Contraceptives (years)* | | 0.01 | 0.01 | | | 0.01 |
| *STDs: Hepatitis B* | | 0.01 | | 0.02 | | 0.01 |
| *Smokes (packs/year)* | 0.01 | | 0.01 | | 0.01 | |
| *Dx* | | 0.01 | 0.01 | | | 0.01 |
| *STDs: HPV* | 0.01 | | | 0.01 | | 0.00 |
| *STDs: Number of diagnoses* | 0.01 | | 0.01 | | 0.01 | |
| *STDs:vulvo-perineal condylomatosis* | 0.01 | | 0.01 | | 0.01 | |
| *STDs: pelvic inflammatory disease* | | 0.01 | | 0.00 | 0.03 | |
| *STDs* | 0.01 | | 0.01 | | 0.00 | |
| *STDs (number)* | 0.00 | | 0.00 | | 0.01 | |
| *STDs:condylomatosis* | | 0.00 | | 0.00 | 0.00 | |
| *Smokes (years)* | 0.00 | | 0.00 | | 0.01 | |
| *Age* | 0.01 | | 0.01 | | 0.01 | |
| *IUD (years)* | 0.00 | | 0.00 | | 0.00 | |
| *Num of pregnancies* | 0.01 | | | 0.00 | 0.01 | |
| *STDs: vaginal condylomatosis* | 0.00 | | | 0.00 | 0.00 | |
| *First sexual intercourse* | | 0.01 | | 0.01 | | 0.01 |
| *Smokes* | | 0.00 | | 0.00 | | 0.00 |
| *Number of sexual partners* | | 0.00 | 0.01 | | | 0.00 |
| *Dx: CIN* | 0.00 | | 0.00 | | 0.00 | |
| *STDs: genital herpes* | | 0.00 | | 0.01 | | 0.02 |
| *IUD* | 0.00 | | 0.00 | | 0.00 | |
| *STDs: syphilis* | 0.00 | | | 0.00 | 0.00 | |
| *STDs:molluscum contagiosum* | | 0.00 | | 0.00 | | 0.01 |
| *Hormonal Contraceptives* | | 0.00 | 0.00 | | | 0.01 |
| *STDs: cervical condylomatosis* | 0.00 | | 0.00 | | 0.00 | |
| *STDs: AIDS* | 0.00 | | 0.00 | | 0.00 | |
| | Second Case | | | | | |
| **Features Importance** | **Idx =10** | | **Idx =12** | | **Idx =15** | |
| | **0** | **1** | **0** | **1** | **0** | **1** |
| *behavior_sexualRisk* | 0.00 | | 0.00 | | 0.00 | |
| *behavior_eating* | | 0.04 | | 0.02 | | 0.03 |
| *behavior_personalHygine* | | 0.02 | | 0.01 | 0.01 | |
| *intention_aggregation* | 0.13 | | 0.13 | | 0.10 | |
| *intention_commitment* | | 0.15 | | 0.12 | 0.13 | |
| *attitude_consistency* | | 0.05 | | 0.02 | | 0.06 |
| *attitude_spontaneity* | 0.05 | | 0.06 | | | 0.02 |

**Table 6** (*continued*)

| Features Importance | First Case | | | | | |
|---|---|---|---|---|---|---|
| | Idx =100 | | Idx=120 | | Idx=150 | |
| | **0** | **1** | **0** | **1** | **0** | **1** |
| *norm_significantPerson* | 0.20 | | 0.18 | | 0.21 | |
| *norm_fulfillment* | 0.06 | | 0.14 | | 0.06 | |
| *perception_vulnerability* | 0.23 | | 0.08 | | | 0.07 |
| *perception_severity* | 0.42 | | 0.44 | | 0.23 | |
| *motivation_strength* | 0.16 | | 0.16 | | 0.01 | |
| *motivation_willingness* | | 0.04 | | 0.02 | | |
| *socialSupport_emotionality* | 0.12 | | 0.09 | | | 0.17 |
| *socialSupport_appreciation* | 0.04 | | 0.02 | | | 0.01 |
| *socialSupport_instrumental* | | 0.03 | | 0.05 | 0.03 | |
| *empowerment_knowledge* | 0.16 | | 0.12 | | | 0.21 |
| *empowerment_abilities* | 0.14 | | 0.07 | | | 0.04 |
| *empowerment_desires* | 0.08 | | | 0.03 | | 0.13 |

that it has a lower training_time_ms of "153" microseconds and a lower error of "0.0717." Most importantly, illustrates the relative weights of the model's most critical variables, the importance contributed to each instance's features using a SHAP summary plot, and the marginal impact of a variable on the outcome using a partial dependence plot (PDP) before model prediction. This, H2O AutoML requires application software such as 'java' to perform simultaneous ML tasks on clusters. While scaling the model, delivers greater efficiency and flexibility. H2O is fast because of these clusters. AutoML is meant to use as few parameters as feasible during modeling to be easily implemented in healthcare organizations. The rest of the procedure is automated to determine the optimal model for this cervical cancer dataset. AutoML is notable for its ability to choose and construct high-accuracy ensemble models. H2O Driverless AI allows researchers to automate ML procedures, allowing physicians to work more quickly and efficiently. This model explained the more complex model's predictions locally by applying the explanation Throughout the process, AutoML trains several Stacked Ensemble models. The exclude_algos option can be used to turn off specific algorithms (or groups of algorithms).

To investigate and further analyze, the AutoML models, use the H2O Model Explainability interface, which can help to decide which model to choose. This model predicts the output directly on complex data. LIME visualization technique helps to explain each prediction. The concept that every complex model is linear on a local scale underpins LIME's operations, as does the assertion that a simple model can be fitted around a single observation to imitate how the global model behaves at that point. The simplest way to interpret the results, though, is to visualize them. LIME provides several plotting tools, more specifically for tabular data. Plot_features is the most essential that generates a visualization with a separate plot for each observation. Three different prediction probabilities have been carried out in this project, that take different feature values such as age, the number of sexual partners, first sexual intercourse, more specifically the number of pregnancies, and others. In the first, second, and third idx values, our model predicts

0.04, 0.08, and 0.04 percent for class '1', whereas 0.96, 0.92, and 0.96 for class '0' in the first case; 0.0 for class '1' and 1.0 for class '0' in the second case respectively. Besides that, we have designed Table 7, which represents the comparative analysis of our proposed method result with previous studies that occurred in the cervical cancer area. This shows our proposed method outperforms with a higher ROC ROC AUC score of 0.974 than other methods recently used in previous studies.

## CONCLUSIONS

AutoML automates the majority of the preprocessing processes in an ML pipeline with minimal human intervention and without sacrificing performance. Moreover, the H2O platform helps physicians to experiment with a variety of methodologies and generate models in a short amount of time. When LIME and AutoML are combined, an interpretable representation is created that is trained on minor deviations of the cervical data. It also aids in the discovery of the smallest set of characteristics that has the best chance of matching the model's prediction on this cervical cancer dataset. LIME can also be used in healthcare companies as unstructured tabular data, with columns representing features and rows representing specific incidents. The prediction object is a three-column data frame in this case. The first column is a class prediction, while the remaining columns are probabilities. LIME employs the explain_instance () method to predict the likelihood of cervical cancer based on thirty-five variables.

H2O AutoML automates the ML processes efficiently within less time but still needs to make quick informative decisions in healthcare firms, in finding the best predictive model. Although AutoML is a positive step toward the widespread adoption of ML technology, there is still a need for ML approaches to entirely replace all required preprocessing procedures in healthcare sectors. Even though GBM is the best approach, our research demonstrates that for three of the targets, both H2O AutoML and LIME outperformed deep learning, and could provide a quick and user-friendly alternative to manual model creation in the future.

### Funding
The authors received no funding for this work.

### Competing Interests
The authors declare there are no competing interests.

### Author Contributions
- Sashikanta Prusty conceived and designed the experiments, performed the experiments, analyzed the data, performed the computation work, prepared figures and/or tables, authored or reviewed drafts of the article, and approved the final draft.
- Srikanta Patnaik performed the experiments, analyzed the data, authored or reviewed drafts of the article, and approved the final draft.

Prusty et al. (2024), *PeerJ Comput. Sci.*, DOI 10.7717/peerj-cs.1916

**Table 7 Comparative analysis of different models in previous studies on cervical cancer research area.**

| Author | Dataset repository | Datatype | Purpose | Method | Target | Dimensions | ROC AUC |
|---|---|---|---|---|---|---|---|
| *Parikh & Menon (2019)* | UCI Machine | Categorical | To predict cancer based on numerous factors | Machine Learning | Biopsy | 858*36 | 0.822 |
| *Fernandes et al. (2018)* | UCI Machine | Categorical | To implement fully supervised optimization of dimensionality reduction and classification models | Machine Learning | Biopsy | 858*36 | 0.687 |
| *Singh & Sharma (2019)* | UCI Machine | Categorical | To classify cancer class | M6 and Decision Tree | Biopsy | 858*36 | 0.779 |
| *Liu, Lu & Lu (2021)* | Kaggle | Categorical | H2O to help in identifying the most predictive ML pipeline | Pharm-AutoML | Biopsy | 858*34 | 0.959 |
| *Suguna & Balamurugan (2022)* | Herlev dataset | Image | To provide an efficient water strider algorithm with autoencoder for cervical cancer diagnosis | WSAAE-CCD | Pap Smear | 917*7 | 0.947 |
| *Hou et al. (2022)* | – | Image | To discuss how AI can be used in cervical cancer screening and diagnosis, particularly to improve the accuracy of early diagnosis | T2WI and Decision Tree | Biopsy | 137*5 | 0.847 |
| *Lilhore et al. (2022)* | China health center dataset | Categorical | To determine the importance of cervical cancer screening factors for classifying high-risk patients | Boruta analysis | Biopsy | 858*36 | 0.534 |
| *Ratul et al. (2022)* | UCI Machine Learning | Categorical | To predict early jeopardies of cervical cancer | Machine Learning | Biopsy | 72*19 | 0.933 |
| *Kruczkowski et al. (2022)* | Interferograms | Histological | To provide suitable decisions for doctors in diagnosing cervical cancer | ML and CNN | Cervical intraepithelial neoplasia (CIN) | 210*18 | 0.950 |
| *Chadaga et al. (2022)* | University of California, Irvine | Histological | To produce reliable predictions | CNN | MRI | 1500*8 | 0.950 |
| *Kaushik et al. (2022)* | Cervical Cancer (2020) | Categorical | To predict cervical cancer | ML | Biopsy | 858*36 | 0.965 |
| *Proposed Method* | UCI Machine Learning | Categorical | To design a user-friendly application, for enhancing ML model such as automatic training and tuning of multiple models within a specified timeframe | H2O AutoML-LIME | Biopsy | 858*30 | 0.974 |

- Sujit Kumar Dash analyzed the data, authored or reviewed drafts of the article, and approved the final draft.
- Sushree Gayatri Priyadarsini Prusty conceived and designed the experiments, performed the experiments, analyzed the data, prepared figures and/or tables, authored or reviewed drafts of the article, and approved the final draft.
- Jyotirmayee Rautaray analyzed the data, authored or reviewed drafts of the article, and approved the final draft.
- Ghanashyam Sahoo analyzed the data, authored or reviewed drafts of the article, and approved the final draft.

## Data Availability

The raw data is available in the Supplemental Files.

The Cervical Cancer Risk Classification dataset is available at https://archive.ics. uci.edu/dataset/383/cervical+cancer+risk+factors and also available at Kaggle: https: //www.kaggle.com/datasets/loveall/cervical-cancer-risk-classification?resource=download.

## Supplemental Information

Supplemental information for this article can be found online at http://dx.doi.org/10.7717/ peerj-cs.1916#supplemental-information.

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
