# Peer review of "Predicting cervical cancer risk probabilities using advanced H20 AutoML and local interpretable model-agnostic explanation techniques"

_PeerJ Computer Science, doi:10.7717/peerj-cs.1916_

## Round 0.1 · original submission · Major Revisions

A series of issues are still present in the article, so it cannot be accepted for publication. The authors should address the points described by the reviewers and prepare a new version of the manuscript.

**Language Note:** The review process has identified that the English language must be improved. PeerJ can provide language editing services - please contact us at [email protected] for pricing (be sure to provide your manuscript number and title). Alternatively, you should make your own arrangements to improve the language quality and provide details in your response letter. – PeerJ Staff

Reviewer 1 ·

Basic reporting

There are a few typos:

Line 26: Please replace "Cancer positioning a major disease" with "Cancer is positioned as a major disease."

Line 33-34: Please replace "the traditional ML technique handles" with "traditional ML techniques handle."

Line 87: Ensure the statement "(WHO)" is formatted correctly as a citation or note. The parentheses are hard to understand.

Line 95: Please replace "machine ML algorithms" with either "ML algorithms" or "machine learning algorithms"

Experimental design

Methods Section of the paper needs some improvements.
What are the features for the dataset? Please give a few examples or explain in plain words what the features typically represent.

In Materials & Methods, please cite the github or website page of the open source program h2o.ai. If you meant the h2o.ai website, then the model is trained on the cloud, please indicate the hardware environment of the cloud cluster.

Validity of the findings

The paper highlights the LIME method for model explanation. Please give examples on the explanation result and the findings behind them.
The authors mentioned the model outperforms previous studies occurred in cervical cancer research, please explain on what aspect the model purposed in the paper is better, introduce what are these methods, and run experiments for comparision.

Reviewer 2 ·

Basic reporting

The manuscript currently references outdated cancer statistics which are at least 5 years old. Updating the paper with the most recent data on cancer statistics is recommended to maintain the relevance and accuracy of the information presented.

The use of "however" in line 69 does not align well with the context. A careful review of the grammar and spelling throughout the manuscript will enhance its readability. Moreover, the introduction contains repetitive statements, which should be refined to avoid redundancy.

The coherence between paragraphs needs to be improved. Presently, there is a lack of clear demarcation between the introduction and literature review sections, resulting in an overlap of information. It is advised to clearly delineate the boundaries by reserving epidemiological data, statistics, and screening issues for the introduction, and focusing on comparing traditional approaches with ML & DL models in the literature review

Experimental design

The data set derived from Kaggle is originally from the UCI Repository, a fact that should be acknowledged to give proper credit to the primary source.

The methodology section lacks clarity in explaining how the automated tool for prediction was utilized. Additionally, the paper seems to lack innovative elements, as even the model selection process is automated. It would benefit the study to introduce more depth and originality in the approach,

The usage of the DX:cancer column in the training phase raises concerns about the validity of the predictions, especially when the objective is to predict the necessity of a biopsy. It is advisable to reassess the inclusion of this variable in the initial training to maintain the integrity of the predictions.

Validity of the findings

The claim regarding the screening ability of the model is contentious given the nature of the dataset utilized. Traditional cervical cancer screening entails processing image data, a component conspicuously missing in the current approach which instead focuses on assessing the risk of developing cancer. The manuscript should revisit this claim to align the narrative with the actual capabilities of the developed model.

The title "Integrated modeling approach to predict cervical cancer in women" seems to overstate the capabilities of the model given that it primarily assesses the risk factors rather than directly predicting cervical cancer occurrences. It would be prudent to modify the title to accurately reflect the scope and the objective of the study.

---

## Round 0.2 · Major Revisions

Some issues raised by the reviewers were correctly addressed by the authors, but some problems still remain.

In particular, if this study is a binary classification project, only binary classification metrics should be employed (MCC, F1 score, accuracy, sensitivity, specificity, precision, NPV, ROC AUC, PR AUC, etc) and not the regression analysis metrics such as RMSE.

Moreover, the authors show results in only one single dataset; the analysis should be repeated on at least one alternative validation cohort dataset. Other datasets on cervical cancer EHRs could be found on Google Dataset Search, re3data.org, Zenodo, Kaggle, FigShare, UC Irvine ML Repository, and other resources.

---

## Round 0.3 · Minor Revisions

The authors added tests on additional datasets but completely ignored my previous comment:

> 1. In particular, if this study is a binary classification project, only binary classification metrics should be employed (MCC, F1 score, accuracy, sensitivity, specificity, precision, NPV, ROC AUC, PR AUC, etc) and not the regression analysis metrics such as RMSE.

Please add results measured through the indicated metrics.

---

## Round 0.4 · Minor Revisions

The results of the binary classification measured through MCC, F1 score, accuracy, sensitivity, specificity, precision, and NPV are still missing. The authors should include them in Table 5 and discuss them in the text.

Plus, what the authors call "AUC" should be called "ROC AUC" or "AUROC".

---

## Round 0.5 · accepted · Accept

The authors correctly addressed my last requests and therefore I can recommend this article for acceptance and publication.